# Preconception care uptake and risk factors for adverse pregnancy outcomes among pregnant women in Tigray, northern Ethiopia: A community-based cross-sectional study

Gebremedhin Gebreegziabher Gebretsadik[1,2*], Andargachew Kassa Biratu[3],
Alemayehu Bayray Kahsay[2], Amanuel Gessessew[4], Zohra S. Lassi[5,6],
Hailemariam Segni[7], Afework Mulugeta[2]

**1** College of Medicine and Health Sciences, Adigrat University, Adigrat, Ethiopia, **2** School of Public Health, College of Health Sciences, Mekelle University, Mekelle, Ethiopia, **3** School of Public Health, College of Medicine and Health Sciences, Hawassa University, Hawassa, Ethiopia, **4** School of Medicine, College of Health Sciences, Mekelle University, Mekelle, Ethiopia, **5** School of Public Health, Faculty of Health and Medical Sciences, University of Adelaide, Adelaide, Australia, **6** Robinson Research Institute, University of Adelaide, Adelaide, Australia, **7** JSI, Addis Ababa, Ethiopia

* gebremedhingebretsad@gmail.com

## Abstract

### Background

Adverse pregnancy outcomes continue to pose a significant global public health challenge, especially in low- and middle-income countries. Although preconception care (PCC) interventions are advised to address this problem, their adoption remains inadequate, supported by scarce evidence particularly in conflict-impacted areas such as Tigray, Ethiopia, where rates of poor outcomes like neural tube defects are notably higher than in other regions. This study investigates the experience of pregnant women regarding the use of PCC in the Tigray, northern Ethiopia.

### Methods

A community-based cross-sectional study was conducted from July 31 to August 16, 2024, involving 764 pregnant women in their first or second trimester. Participants were consecutively enrolled from clusters until the predetermined sample size was achieved. Data were collected through interviewer-administered questionnaires in accordance with World Health Organization, and Centers for Disease Control and Prevention, and national guidelines. PCC uptake was measured as the receipt of any service component (screening, counseling, or management) during healthcare consultations. We used SPSS version 27.0 to analyze PCC uptake and its associated factors. Descriptive and binary logistic regression statistics were used in the analysis. Finally, data was presented using text, tables, and figures as appropriate.

**Data availability statement:** All relevant data are within the manuscript and its Supporting information files.

**Funding:** The author(s) received no specific funding for this work.

**Competing interests:** The authors have declared that no competing interests exist.

**Abbreviations:** ANC, Antenatal Care; APOs, Adverse Pregnant Outcomes; CDC, Centers for Disease Control and Prevention; EDHS, Ethiopia Demographic Health Survey; HEP, Health Extension Programme; PCC, Preconception Care; SDG, Sustainable Development Goal; SSA, Sub-Saharan Africa; WHO, World Health Organization; WDGs, Women Development Groups.

## Results

In this study, the overall uptake of PCC services was 7.2%. All participants in the current pregnancy were exposed to at least one risk factor for adverse pregnancy outcomes. Factors such as women's decision-making power, having information about PCC, HIV screening during the current pregnancy, and perceived susceptibility to preconception risks showed a statistically significant positive association with the uptake of PCC services.

## Conclusion

The uptake of PCC services was very low. Addressing the low uptake of PCC services requires a multifaceted strategy, including public health campaigns via media and social forums, strengthened health extension programs, and the integration of a reproductive life plan tool to improve health-seeking behavior among women.

## Introduction

Adverse pregnancy outcomes (APOs) represent a significant global public health burden, with a disproportionate impact in Sub-Saharan Africa(SSA) [1,2]. Key risk factors, including malnutrition, chronic disease, and substance use, drive these poor outcomes [3,4]. According to the World Health Organization (WHO) and Centers for Disease Control and Prevention (CDC), preconception care (PCC) offers a proven, cost-effective intervention to identify and mitigate such risks through comprehensive assessment, health education, and tailored management [5,6]. It is considered a component of the maternal continuum of care; however, it is not commonly practiced [7]. Based on 81 countdown countries to achieve the Sustainable Development Goals (SDGs) by 2030, the rate of decline in the prevalence of stillbirths and maternal and child mortality needs to accelerate considerably, which is possible with a rapid scale-up of effective interventions like PCC [8]. PCC is highly valued and widely accepted for preventing and reversing adverse health outcomes in mothers and infants [9]. For instance, it has been shown to reduce birth defects from 5.6% to 2.5% and lower the risks of preterm birth, neonatal complications, and maternal complications by 70%, 54% and 60% respectively. Additionally, it enhances antenatal care services [10].

In Ethiopia, the prevalence of APOs ranges from 18.3% to 32.5% [11–13], with key risk factors that lead to APOs including alcohol consumption(68.7%), khat chewing(27.6%), smoking (20.3%), and underweight (36.2%) [14]. Additionally, folate deficiency is highly prevalent, affecting 84% of women of reproductive age [15], and 49.3% of pregnant women [16]. As a result, according to the WHO (2020) report, Ethiopia is among the top 10 countries with the highest newborn mortality rates [17].

This situation is profoundly worsened in the Tigray region due to recent conflict, which has devastated the health system and precipitated sharp increases in gender-based violence(GBV) affecting 43.3% of women of reproductive age [18], with

many infected with sexually transmitted infections (STIs), including HIV [19]. Additionally, 60.1% of households experienced moderate or severe hunger [20]. The prevalence of neural tube defects (NTDs) rose to 262.5 per 10,000 births [21] compared to 131 per 10,000 before the war [22]. Maternal mortality also surged to 840 deaths per 100,000 live births during the conflict as compared with the pre-war level of 266/100,000 [23].

Despite its proven benefits, PCC implementation varies globally [24], with utilization rates of 42.2% in China [25], 83% in Belgium [26], and 51% in the UK [27]. Moreover, PCC uptake among women of childbearing age in SSA stands at 24.05% [28], with varying proportions, indicating the limited implementation of the services in the region's healthcare system. Previous studies from Ethiopia unequivocally demonstrated the importance of PCC in decreasing health care cost, maternal and neonatal mortality, stillbirth, and unplanned pregnancy [29,30]. Nevertheless, its uptake remains low [31]. Moreover, the very ambitious Health Sector Transformation Plan II (HSTP II) of Ethiopia that targets the reduction of neonatal mortality from 33 to 21 per 1,000, and maternal mortality from 401 to 271 per 100,000 live births [32] reaffirms the need for implementation of PCC to this effect.

Although the Ethiopian government has prioritized PCC since adopting a national policy in 2020 [32], supporting implementation guidelines [33], and early progress is promising, indicated by a study finding that 53.54% of healthcare providers demonstrate proficient PCC practice [34], the uptake of PCC remains low [31]. While early progress is encouraging, the program's implementation faces severe challenges. The initiative has been particularly compromised in the Tigray region, where recent conflict has weakened the health system and diverted priorities away from newer programs like PCC [35].

Research on PCC implementation in Ethiopia is limited, and further studies are necessary to gain a deeper understanding. The intervention was in its infancy, making it challenging to assess its progress [36]. Furthermore, paucity exists on the understanding of women's experiences with pregnancy preparation, the components of PCC interventions, and risk factors for APOs, particularly in conflict-affected areas. To address this gap, we conducted a community-based survey in both urban and rural areas to evaluate pregnancy preparation among pregnant women, including the content of PCC interventions and the risk factors for APOs.

## Methods and materials

### Study setting

Tigray, one of Ethiopia's regions, is located in the northernmost part of the country. According to the 2024 annual performance report for the health sector in Ethiopia, its population is estimated at approximately 5.9 million, comprising 2.9 million males and 3 million females. The region is further divided into seven zones, which are then subdivided into smaller administrative districts. Based on the study conducted in the region, 54.9% of pregnant women had attended at least one antenatal care visit, while only 20.1% had received optimal antenatal care [37]. The study was conducted in six districts in the Tigray region, including Ahferom, rural Adwa, and Adwa town in the Central Zone, and Kilite-Awlaelo, Tsirae wemberta, and Wukro town in the Eastern Zone of Tigray, Ethiopia. Together, these districts/ towns encompass 68 villages or kebelle and a population of 549,419, with an estimated 18,680 pregnant women. A total of 30 kebelle/tabias were included in the present study (Fig 1).

### Study design and population

We conducted a community-based cross-sectional survey of pregnant women in their first and second trimesters, as determined by the self-reported reliable first day of their last menstrual period and ultrasound assessments performed before data collection. We implemented several methodological adaptations during data collection, including hiring local guides, conducting ongoing security assessments, and deploying data collectors in teams to minimize bias and enhance access to remote communities in the post-conflict setting.

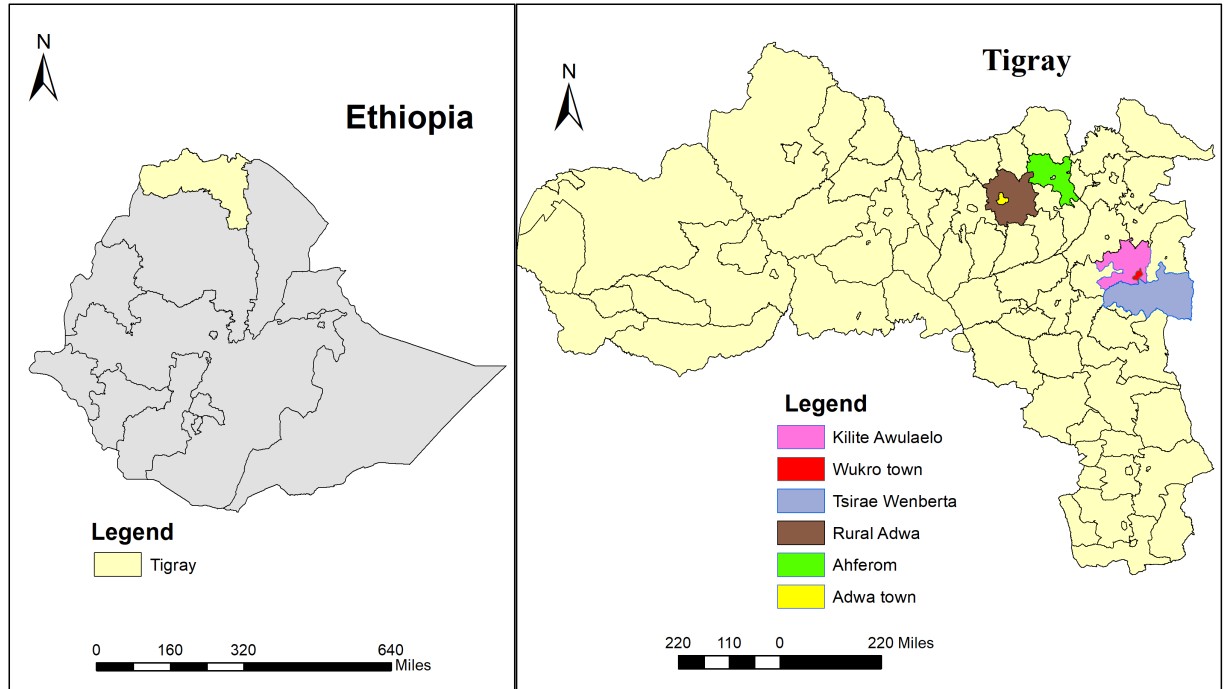

**Fig 1. Study sites for the PCC uptake study among pregnant women from Tigray, Ethiopia, 2024.**

## Inclusion and exclusion criteria

Eligible participants were pregnant women in their first or second trimester, permanent residents of the selected Tigray study areas for at least six months, who provided informed consent. We excluded women who were critically ill or unable to communicate.

## Identification of pregnant women

The list of pregnant women was registered in each kebelle or tabia, and the Health Extension Workers (HEWs), in coordination with women development groups (WDGs), maintained a list of registered pregnant women. Following the conflict in selected districts, food distribution was provided to pregnant and lactating mothers with a MUAC of less than 23 cm, supported by various partners. While confirming pregnancy status does not guarantee access to antenatal care (ANC) services, most women who suspected they were pregnant checked their status at a health facility earlier than before. They received a slip from the facility verifying their pregnancy, which they then provided to the HEWs. As a result, nearly all women were aware of their pregnancy status before the survey. Furthermore, before data collection, the Tigray Regional Health Bureau, in collaboration with the WHO, conducted a SMART survey at the regional level. HEWs were involved in this survey to identify pregnant women and lactating mothers, creating a comprehensive profile to inform intervention efforts. This survey provided an opportunity to identify women in their first and second trimesters for the study.

## Sample size and sampling techniques

The sample size was calculated using a two-sided Z-test for proportions in Epi-Info software, with 80% power and a 5% significance level. Based on a previously reported uptake of PCC proportion of 16.27% [38] and an assumed 10% increase post-intervention, an initial calculation was performed. This figure was then adjusted upward by a design effect

of 2.5 to account for the cluster sampling design and by 20% to accommodate potential non-response [39], yielding a final minimum sample size of 778 participants for the baseline study. A multi-stage cluster sampling approach was employed. Thirty kebeles/tabias were randomly selected from a total of 68 across two zones. Participants were allocated proportionally to each selected cluster based on population size. Within each cluster, all registered pregnant women in their first or second trimester were consecutively enrolled until the predetermined sample size was achieved.

## Data collection methods and measurements

After identifying eligible pregnant women, we adapted a questionnaire from literature sources, including published research articles, EDHS, WHO survey tools, and the CDC [6,40–44]. Experienced data collectors with public health officers, midwives, and nurses with BSc, MSc, and MPH degrees were included in the data collection to gather data via the Kobo Collect Smart tool using tablets. Three days of training were provided for data collection and supervisors on all questionnaire components, including MUAC measurement, ethical principles, and the study's purpose. We pretested the questionnaire with 39 pregnant women in a different zone to assess its clarity, relevance, and cultural appropriateness. After incorporating their feedback, we uploaded the final version to KoBoToolbox. From July 31 to August 16, 2024, the trained healthcare providers collected community-based data through an interviewer-administered survey using a pre-tested tool, which included MUAC measurements. We used Kobo Smart Survey software and collaborated with the Tigray Regional Health Bureau and the local health district to gather the following data:

**Access to health services:** The time needed to reach the nearest health facility was measured to assess access to health services, with a travel time of 30 minutes or less indicating better access at the time of inclusion [45].

**Preconception care:** The WHO convened a collaborative meeting with researchers, healthcare practitioners, programme managers, UN agencies, and partner organizations to establish a global consensus on PCC as a vital strategy for reducing maternal and child mortality and morbidity. The discussions emphasized the need for a standardized universal PCC package, complemented by tailored interventions to address local health priorities [5]. For this study, we developed a comprehensive pre-pregnancy care framework by integrating both the WHO's basic and extended PCC packages [5] with Ethiopia's national preconception care guidelines [33]. The components of PCC package includes; family planning and contraception; nutritional support; folic acid supplements; screening, counselling and management for chronic medical illness; reducing substance use (stopping tobacco and alcohol, and cutting back on coffee); encouraging physical activity; checking for reproductive organ problems and cervical cancer; providing sexual health services (like screening, counselling, and treatment) and preventing gender-based violence; protecting against vaccine-preventable diseases; assessing genetic risks; offering dental health services; controlling infectious diseases(HIV and STIs); preventing female genital mutilation; reducing environmental health risks (like indoor air pollution); providing mental health services; and managing medications that could negatively affect pregnancy outcomes [5,6,27].

**Measurement of PCC service uptake:** The uptake of PCC was measured by determining whether women had consulted a healthcare provider for PCC and had received at least one service from the WHO-recommended package and national PCC guideline before pregnancy [5,33,31]. Uptake was categorized as follows: (1) no uptake (no components received); (2) partial uptake (at least one component received); and (3) optimal uptake (folic acid supplementation plus at least one other component) [27,33]. According to guidelines, women with childbearing potential should receive at least one dedicated PCC visit [6], with additional visits as needed based on individual risk factors.

**Pregnancy preparation measurement:** Pregnancy preparation was assessed by asking: *"What preparation or actions did you take before your current pregnancy?"* Response options included: No preparation, self-prepared at home, or consulted healthcare providers.

We measured awareness of PCC with a single question: "Have you ever heard about pre-pregnancy care?" Women who answered "yes" were considered to have information about PCC.

To assess attitudes toward preconception care, we developed a validated 12-item Likert scale questionnaire through a literature review, expert consultations, and a pilot study. The scale incorporated reverse-scored statements, ensuring a higher total score (range: 12–60) consistently indicated a more positive attitude. Based on their scores, respondents were dichotomized into two groups: those with a "positive attitude" (scores ≥ the mean) and those with a "negative attitude" (scores < the mean) [46,47].

**Community health services:** We assessed the implementation of the health extension package (HEP) by verifying whether the women's households were certified as model households at the time of inclusion. A model household was defined as one that received short-term training on the HEP and subsequently implemented it [48,49]. Being a household model (MHH): a family that implements all the components of HEP and has received certificates of appreciation from responsible bodies [50].

**Maternal health services characteristics:** Obstetric ultrasound services were assessed by determining if participants received at least one ultrasound service within the first 24 weeks of gestation [51]. Cervical cancer screening status was defined by whether participants had undergone screening at least once in the past five years (yes/no). Preconception vaccination status was determined by asking if participants received tetanus diphtheria (Td1–Td4), COVID-19, HPV vaccine, and Hepatitis B vaccines before the current pregnancy (yes/no). Fasting status was measured by asking if participants abstained from animal-based foods (meat, dairy, eggs) for religious reasons on Wednesdays, Fridays, and during extended fasting periods (yes/no) [52]. This study assessed women's decision-making power based on their involvement in financial decisions for major household purchases, decisions regarding visits to family or relatives, and decisions about their own healthcare. Responses were coded on a 3-point scale: 2 ("mainly wife"), 1 ("joint with husband"), or 0 ("mainly husband or someone else"), with total scores ranging from 0 to 6. Women who made decisions either independently or jointly with their husbands were classified as having decision-making power [53]. Both perceived susceptibility and perceived severity were classified as low, moderate, or high based on their respective scores [54].

## Risks factors for adverse pregnancy outcomes

We classified participants as exposed to a risk factor for poor pregnancy outcomes if they had at least one of the risk factors listed below.

**Substance use:** We assessed the status of medication use through the following categories: self-medication with herbal or natural remedies, over-the-counter medications (such as aspirin), weight-loss medications, athletic products, drugs, and recreational drugs/illicit substances. We assessed coffee intake by asking, "Do you currently drink coffee?" Respondents answered "yes" or "no," and those who answered "yes" were further categorized by daily intake as either fewer than three cups or three or more cups during the current pregnancy [55]. We assessed intake of alcohol by asking "did you drink any amount of alcohol during pregnancy?" report when participants respond "yes"

**Reproductive health and obstetrics-related risks:** Subfertility was defined as the inability to conceive within 6 months of attempting pregnancy for women over 35 years of age and within 12 months for women under 35 years [56,57]. We assessed the pregnancy interval by asking for the number of months between the current and previous pregnancies, categorizing intervals below 24 months or above 59 months as abnormal [58]. We assessed pregnancy planning using the London Measure of Unplanned Pregnancy (LMUP), a six-item questionnaire with a score range of 0–12. Scores of 0–3 indicate "unplanned," 4–9 indicate "ambivalent," and 10–12 indicate "planned [59]. The history of childbirth was categorized as follows: Grand multi-parity is defined as a parity of 5 or more deliveries [60]. APOs are adverse outcomes of both fetal and maternal outcomes, and they include at least one of the following: stillbirth, neonatal death, low birth weight, preterm birth, congenital anomalies, perinatal death, miscarriage, eclampsia, pre-eclampsia, small for gestational age, Antepartum hemorrhage, postpartum hemorrhage, recurrent abortion, pregnancy induced hypertension, and gestational diabetic mellitus [13].

**Nutrition-related risks:** Mid-upper arm circumference (MUAC) is a simple, reliable alternative to BMI for assessing nutritional status, already applied in resource-limited settings and supported by evidence in pregnant women [61].

Pregnant women with a MUAC of ≥ 23 cm were classified as well-nourished, and those with MUAC < 23 cm as undernourished [62,63]. Trained healthcare professionals measured MUAC by averaging three readings taken midway between the shoulder and elbow of the left arm, with the arm hanging freely [64]. Minimum Dietary Diversity for Women (MDD-W) was assessed using a 24-hour dietary recall method per FAO guidelines. This involved asking women whether they had consumed any items from ten predefined food groups in the preceding 24 hours. A woman was classified as having adequate dietary diversity if she had consumed items from five or more groups. This method is an internationally recognized, cost-effective tool designed for rapid population-level screening of micronutrient deficiency risk [65–67].

The short form of the International Physical Activity Questionnaire was used to collect data on the types, frequency, and duration of physical activities that women engaged in over the past week [68,69]. The categorized activity levels as low, moderate, or high were derived by calculating weekly MET-minutes, where MET values (walking = 3.3, moderate = 4, vigorous = 8) were multiplied by the duration and frequency of each activity. For example, walking 30 minutes for 5 days yields 495 MET-minutes. Classification was then determined as follows: high activity (≥1500 MET-minutes of vigorous activity on ≥3 days, or ≥3000 MET-minutes from any combination), moderate activity (≥600 MET-minutes, or meeting specific frequency and duration thresholds), and low activity (not meeting moderate or high criteria). Finally, we classified participants as 'active' by merging the moderate and high activity groups, and as 'inactive' for the low activity group [70].

**Medical-related risks:** We assessed dental health by asking, "Do you currently have any of the following: red or swollen gums, tender or bleeding gums, pain when chewing, loose teeth (tooth decay), or dental or gum pain?" If a woman answered "yes" to any of these questions, we reported it as having a dental health problem. To assess psychological distress, we used a 10-question scale with responses ranging from 1 to 5, resulting in total scores between 10 and 50. The cut-off scores indicate: 10–19 for no mental disorder, 20–24 for mild disorders, 25–29 for moderate disorders, and 30–50 for severe disorders [71,72]. The level of appetite during the current pregnancy was assessed by asking if the woman experienced anorexia during this time.

**Other risk factors:** The risk of environmental and household exposure was reported when participants answered "yes" to any of the following questions. Do you use an open or traditional stove? Do you use a coal or fuel stove for cooking/ heating? Was there no ventilation during heating or cooking? Do you not use a separate kitchen for cooking? Have you undergone X-ray or radiation therapy? Have you been exposed to organic solvents (chemicals used in industries or installations like benzene, methanol)? Have you been exposed to pesticides (Insecticides, herbicides and fungicides)? And have you had contact with fertilizers? Pregnant women were classified as having a genetic risk if they answered "yes" to any of the following: Do you, your partner, previous children, or other relatives have a birth defect, genetic condition, developmental delay, or learning disability; two or more miscarriages; prior pregnancy ended due to birth defect genetic issues; or an age of 35 or older at delivery? Insufficient sleep as less than 6 hours per night based on the question, "How many hours of sleep do you usually get on a weekday/weekend?" [73]. The hurt, insult, threaten and scream (HITS) is a four-question self-report tool that screens for intimate partner violence (IPV) frequency on a five-point scale from 1 (never) to 5 (frequently). A total score above 10 indicates the presence of violence [74].

## Data quality assurance

Data quality was ensured through a multi-stage process. A comprehensive questionnaire, developed from literature and expert knowledge, was adapted to the local context and pre-tested with 39 pregnant women for refinement. It was then deployed digitally via KoBo Toolbox. To guarantee high-quality data, all field staff underwent extensive training on protocols, MUAC measurement, and the digital tool. Quality was further strengthened by the platform's real-time monitoring, automated validation features (skip logic, constraints), and rigorous on-site supervision to verify questionnaire completeness and consistency.

### Data management and analysis

Descriptive statistics, including proportions, means, and standard deviations, were calculated using SPSS Version 27.0. Besides, we employed Principal Component Analysis (PCA) to measure perceived susceptibility and severity of preconception risks. The variances explained by perceived susceptibility and severity of preconception risks were 69% and 76%, respectively. Reliability analysis showed Cronbach's alpha values of 0.89 for perceived susceptibility and 0.88 for perceived severity. In addition to descriptive statistics, binary logistic regression was used to assess the associations between explanatory variables and PCC uptake. Model fitness was assessed with the Hosmer and Lemeshow test, which was non-significant ($p = 0.153$). Collinearity was also examined with variance inflation factor (VIF) values ranging from 1.01 to 1.11. Statistically significant was set at a p-value of less than 0.05.

### Ethical considerations

The study received approval from the Institutional Review Board of Mekelle University, College of Health Sciences (reference: MU-IRB2075/2023), ensuring that all methods were conducted in compliance with the relevant guidelines and regulations. A support letter to conduct the study was also obtained from the Tigray Health Bureau and local health offices. We attached a one-page consent form to the questionnaire, outlining participant autonomy, study objectives, potential risks and benefits, and confidentiality assurances. Verbally informed consent was obtained from all participants following a comprehensive explanation of the study's purpose, the right to withdraw at any time, and other relevant details. This use of verbal consent was approved by the Institutional Review Board of Mekelle University, College of Health Sciences.

## Results

### Participant characteristics

The study included 764 pregnant women, yielding a 98.2% response rate. Among them, 243 (31.8%) were aged 25–29 with a mean age of 28.3 years (±5.75), and 45 (5.9%) were 19 years or younger. Of the participants, 515 women (67.4%) lived in rural areas. Approximately 318 (41.6%) of participants had access to a radio or TV, while 314 (41.1%) were married before the age of 18. Additionally, 191 (25%) of the participants had to walk 30 minutes or more to reach the nearest health facility (Table 1).

### Uptake of PCC services

Overall, 55 (7.2%) (95% CI = 5.5 to 9.2) of pregnant women had experienced PCC services prior to their current pregnancy (Fig 2), with 20 (2.6%) pregnant women received partial services, and 35 (4.6%) received optimal PCC services.

### Components of PCC interventions

In this study, among the various components, family planning and contraception 44 (80%) and nutritional support 43 (78.2%) were the most frequently received services, while, services related to reducing environmental health risks, dental health services, checking female reproductive organ anomalies and cervical cancer, as well as preventing female genital mutilation, were not utilized (Fig 3).

### Pregnancy preparations

About 85.5% (n = 653) of the women became pregnant without any prior preparation, while 56 (7.3%) made home-based preparations before pregnancy. Only 93 women (12.2%) had information about PCC services, with 79 of them (85%) receiving the information from healthcare providers. Lack of awareness was the most reported reason for not receiving PCC services, as indicated by 226 women (31.9%) (Table 2).

**Table 1. Socio-demographic characteristics of participants in Tigray, Ethiopia, 2024(n = 764).**

| Variables | | Frequency | Percentage |
|---|---|---|---|
| Women's age in years | ≤19yrs | 45 | 5.9 |
| | 20-24yrs | 157 | 20.5 |
| | 25-29yrs | 243 | 31.8 |
| | 30-34yrs | 174 | 22.8 |
| | 35-39yrs | 128 | 16.8 |
| | ≥40 | 17 | 2.2 |
| Residence | Rural | 515 | 67.4 |
| | Urban | 249 | 32.6 |
| Able to read & write | No | 104 | 13.6 |
| | Yes | 660 | 86.4 |
| Women education | No education | 92 | 12 |
| | Primary education | 255 | 33.4 |
| | Secondary education | 331 | 43.3 |
| | College and above | 86 | 11.3 |
| Husband education | No education | 99 | 13 |
| | Primary education (2–71–8) | 257 | 33.6 |
| | Secondary education (10,9–12) | 300 | 39.3 |
| | College and above | 108 | 14.1 |
| Husband age | 20-29yrs | 172 | 22.5 |
| | 30-39yrs | 347 | 45.4 |
| | 40-49yrs | 203 | 26.6 |
| | ≥50yrs | 42 | 5.5 |
| Have radio or TV | No | 446 | 58.4 |
| | Yes | 318 | 41.6 |
| Age at first marriage | <18yrs | 314 | 41.1 |
| | ≥18yrs | 450 | 58.9 |
| Family size | <5 | 628 | 82.2 |
| | ≥5 | 136 | 17.8 |
| Women occupation | Housewife | 558 | 73 |
| | Self employed | 69 | 9 |
| | Government employed | 51 | 6.7 |
| | Farmer | 86 | 11.3 |
| Religion | Orthodox | 730 | 95.5 |
| | Others[a] | 34 | 4.5 |
| Access to a health facility in minutes (one way) | <30 minutes | 573 | 75 |
| | ≥30 minutes | 191 | 25 |

NB: [a] Others (Muslim & catholic)

## Community health, reproductive and obstetric health conditions, and related variables

Five hundred forty (70.7%) women had low decision-making power to obtain healthcare services, make large household purchases, and visit family or relatives. Additionally, only 167 women (21.9%) had information on the model family, 55 (7.2%) attended the pregnant women's forum, and 73 (9.6%) were members of community health insurance. In this study,

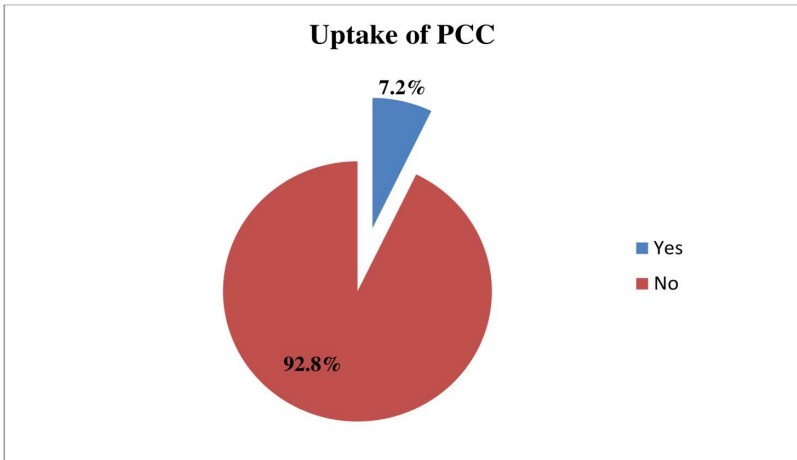

**Fig 2. Percentage of uptake of PCC services among participants in Tigray, Ethiopia, 2024(n = 764).**

71(12.1%) were grand multipara and 155(20.3%) women had at least one history of abortion. Besides, of the 603 women who had previous history of pregnancy 36(4.7%) still births and 60(7.9%) neonatal deaths were observed (Table 3).

**Maternal health services characteristics.** Out of 764, about 615 (80.5%) women had attended at least one ANC visit. Of those, 299 (48.6%) received obstetric ultrasound services before 24 weeks of gestation. The 191(63.9%) women received the obstetric ultrasound services from a private facility. A total of 280 (36.6%) women had never used contraceptives, while 280 (57.9%) women had used contraceptives before their current pregnancy.

The main reasons for not using contraceptives before the current pregnancy were side effects like bleeding 51, 25%) and lack of information 23, 11.8%). Around 455 (59.6%) women were taking iron-folic acid supplements during the current pregnancy. The majority of pregnant women (84.2%, 492) consumed some coffee, and around 85.3% (652) reported fasting during their pregnancy.

In terms of immunization, 266 women (34.8%) had received Td4 or higher vaccines, and 665 women (87%) had received the COVID-19 vaccine. However, only 48 women (6.3%) had received the Hepatitis B vaccine, and 36 women (4.7%) had received the HPV vaccine. Moreover, 22 (2.9%) women reported having cervical cancer screenings in the past five years, 316 (41.4%) women were screened for anemia during pregnancy, and 517 (67.7%) women were tested for HIV (Table 4).

**Risk factors for adverse pregnancy outcomes characteristics.** In the study, all pregnant women were exposed to at least one risk factor for APOs, with some experiencing up to 15 different risk factors. About 546 participants (71.5%) had between 5 and 9 risk factors for APOs (Fig 4).

## Risk factors for adverse pregnancy outcomes

Among all risk factors, the most prevalent were environmental and household exposure 758 (99.2%), poor dietary practice 544(71.2%), unplanned pregnancy 448(58.6%), and alcohol consumption during pregnancy 438(57.4%) (Table 5).

## Factors associated with uptake of PCC services

We calculated the adjusted odds ratio (AOR) to determine the factors associated with the uptake of PCC services. The results showed that pregnant women who had good women decision making power (AOR 5.27, 95% CI 2.85–9.72), had information about PCC (AOR 2.9,95% CI 1.42–5.91), tested for HIV during the current pregnancy (AOR 2.27, 95%

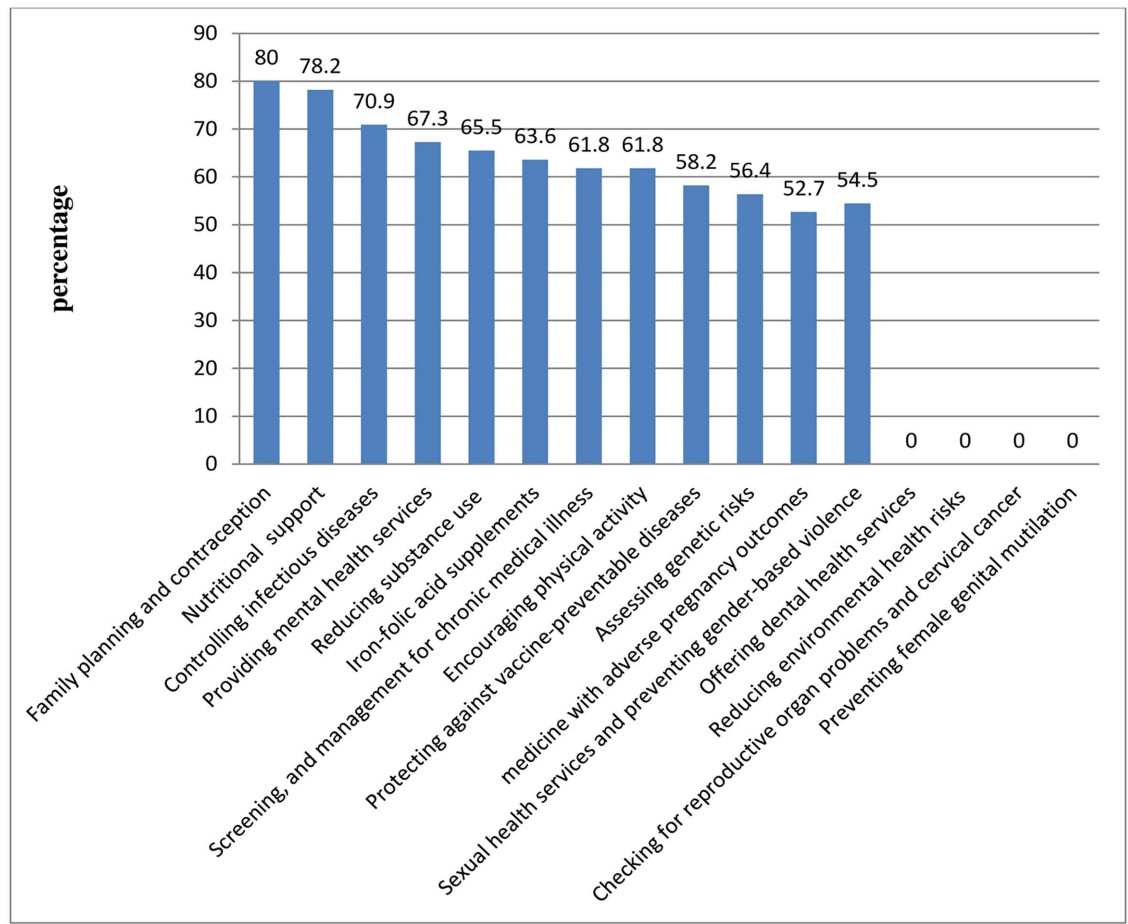

**Fig 3. Proportion of uptake of components of PCC intervention among participants, in Tigray, Ethiopia, 2024 (n = 55).**

CI 1.01–5.11), and had perceived susceptibility to preconception risks (AOR 3.25, 95% CI 1.19–8.85) were statistically significant positive association with uptake of PCC services. Residence, husband occupation, history of medical illness, planned pregnancy, and membership in community health insurance did not show any statistically significant association with uptake of PCC services (Table 6).

## Discusion

This study addresses a missed component (PCC) of the continuum of maternal care services in northern Ethiopia. Limited research has been conducted on PCC services in Ethiopia, emphasizing the need for further studies on their uptake [36]. Hence, this study assessed the experience of women's pregnancy preparation, including the uptake of content of PCC interventions and their risk factors for APOs.

The present study revealed low uptake of PCC services with only 7.2% of pregnant women utilizing them. Additionally, all pregnant women were exposed to at least one risk factor for poor pregnancy outcomes. Moreover, women's decision-making power, information about PCC, HIV testing during the current pregnancy, and perceived susceptibility to preconception risks were statistically significantly associated with PCC service uptake.

**Table 2.  Uptake of PCC services among pregnant women from Tigray, Ethiopia, 2024 (n = 764).**

| Variables | Categories | Frequency | Percentage |
|---|---|---|---|
| Actions made before current pregnancy (n = 764) | Nothing do | 653 | 85.5 |
| | Home-based preparation | 56 | 7.3 |
| | Consult HCPs[a] | 55 | 7.2 |
| Uptake of PCC | No | 709 | 92.8 |
| | Yes | 55 | 7.2 |
| Where did you receive PCC services? | Health center | 19 | 34.5 |
| | Governmental Hospital | 24 | 43.6 |
| | Private clinic | 12 | 21.8 |
| Reason for not seeking PCC (n = 709) | Respondent didn't think it necessary | 106 | 15 |
| | Unavailability of the services | 105 | 14.9 |
| | Lack of awareness about the services | 226 | 31.9 |
| | Not know where to go | 157 | 22.1 |
| | Unplanned pregnancy | 43 | 6.1 |
| | Lack of money for transport | 36 | 5 |
| Attitude level | Poor attitude | 373 | 48.8 |
| | Good attitude | 391 | 51.2 |
| Informed about PCC | No | 671 | 87.8 |
| | Yes | 93 | 12.2 |
| Source of information (n = 93) | Healthcare providers | 79 | 85 |
| | Others[b] | 14 | 15 |

NB: HCPs[a], HealthCare Providers, Others[b] (Mass media, Family or friend Leaflets/brochures and school/college)

This study revealed an alarmingly low PCC uptake of 7.2% a figure substantially lower than previous reports from Africa [28], Ethiopia [31], and the national 2025 target [32]. Although the importance of PCC in reducing maternal and perinatal morbidity and mortality has been well-documented for a long period [5,6], our findings give clue evidence that PCC receives little attention in Ethiopia. This calls for an integrated effort to increase the very low PCC utilization to attain the ambitious SDG set by Ethiopia.

Although Ethiopia introduced the PCC program in 2020, the war in the Tigray region, which began in November 2020 severely damaged the healthcare system, disrupting maternal care and essential services. As a result, the PCC program struggled to launch effectively, and existing maternal care services faced significant interruptions. Additionally, maternal health efforts have primarily focused on prenatal and skilled birth care, with less emphasis on PCC. To address gaps, stakeholders and the government should work against the clock to introduce the services, implementing effective awareness-raising strategies and integrating PCC into routine maternal health services. Moreover, strengthening community health insurance is a recommended strategy to improve awareness of PCC and uptake [5,6,27].

Evidence suggests that boosting awareness and uptake of PCC through community campaigns and pre-pregnancy visits is a cost-effective strategy to improve preconception health, promote pregnancy planning, prevent birth defects, and manage pregnancy-related complications [5,6,27]. Participants mentioned that the main reasons for the low uptake of PCC services were a lack of awareness (31.9%) and uncertainty about where to access them (22.1%). This study revealed various barriers to getting information about PCC such as being uneducated, lack of access to information outlets such as radio and television, difficulty accessing health facilities, and the fact that the majority are housewives. These factors should be considered in developing a strategy to increase awareness of PCC, which is pivotal to increasing PCC uptake. Contrary to our report of a very low level of information about the services (12.2%) in this article, a previous study

Table 3. Community health variables and reproductive and obstetric conditions of participants in Tigray, Ethiopia, 2024.

| Variables | | Frequency | Percentage |
|---|---|---|---|
| Have you attended a forum for pregnant women? | No | 709 | 92.8 |
| | Yes | 55 | 7.2 |
| Have you ever discussed PCC in the pregnant women's forum? (n=55) | No | 33 | 60 |
| | Yes | 22 | 40 |
| Member of community health insurance | No | 691 | 90.4 |
| | Yes | 73 | 9.6 |
| Received information about PCC from HEWs | No | 691 | 90.4 |
| | Yes | 73 | 9.6 |
| Gravida (n=764) | Primigravida | 161 | 21 |
| | 2-4 gravida | 420 | 55 |
| | Grand multigravida | 183 | 24 |
| Birth (para)(n=587) | Primipara | 169 | 28.2 |
| | 2-4 para | 347 | 59.1 |
| | Grand multipara | 71 | 12.1 |
| Stillbirth (n=603) | No | 567 | 95.3 |
| | Yes | 36 | 4.7 |
| Neonatal death (n=603) | No | 543 | 92.1 |
| | Yes | 60 | 7.9 |
| History of abortion (n=603) | No | 609 | 79.7 |
| | Yes | 155 | 20.3 |

from Ethiopia reported that 69.4% of women had information about PCC [50]. This discrepancy may be attributed to factors such as 67.2% having prior PCC experience with at least one aspect of care, 45.5% being knowledgeable, and 40% reporting the availability of a PCC unit in nearby health facilities [50]. Additionally, differences in study participants, which included mothers who gave birth within the last 12 months, along with potential recall bias and associations with ANC services, may have contributed to inflated figures in previous studies.

As it is a very well-known fact across the globe, war has a devastating negative effect on the health and health system of a country and its society. The war in Tigray was arguably the bloodiest war, which caused destruction to the weak health system of the region [35,75]. Consequently, it significantly hindered advocacy efforts for the new program.

Evidence suggests that enhancing information about PCC requires public campaigns, education through media and social forums, and government-led initiatives, including integrating PCC into healthcare services [76,77]. These efforts were significantly impacted by the two-year war in the region. To avert this unfortunate situation, the government, both at the regional and federal levels, should prioritize awareness, particularly through the help of leveraging community engagement via civic organizations and women's groups for effective outreach. Additionally, hews should counsel women on home-based preparation and use diverse communication strategies to promote healthy lifestyle changes.

The Ethiopian ministry of health recommends routine obstetric ultrasounds for all pregnant women within 24 weeks of gestation to enhance fetal anomaly detection and improve pregnancy outcomes [76]. In this study, 45.4% of pregnant women underwent basic ultrasounds, with 64% conducted in private facilities. This coverage is lower than a facility-based study in Addis Ababa, likely due to differences in study populations, as this research included both rural and urban participants and focused on early gestation. The findings are also lower than those in Nigeria (96.1%) [77] and China (83.5%) [78]. Early detection of congenital anomalies, uterine abnormalities, or ovarian cysts is crucial for improving women's engagement in inter-conception care and risk prevention. Routine ultrasounds support PCC, facilitate reproductive health

**Table 4. Maternal health services among pregnant women from Tigray, Ethiopia, 2024 (n = 764).**

| Variables | | Frequency | Percentage |
|---|---|---|---|
| ANC services | No | 149 | 19.5 |
| | Yes | 615 | 80.5 |
| Received basic obstetrics ultrasound services within 1–24 weeks of GA for the current pregnancy (n = 615) | No | 336 | 54.6 |
| | Yes | 279 | 45.4 |
| Reason for ultrasound services (n = 299) | Routine services | 261 | 87.3 |
| | High-risk pregnancy | 18 | 6 |
| | Emergency | 20 | 6.7 |
| Place of obstetrics ultrasound services received (n = 299) | Health center | 16 | 5.4 |
| | Hospital | 92 | 30.8 |
| | Private health facility | 191 | 63.9 |
| Used contraceptive before current pregnancy (n = 484) | No | 204 | 42.1 |
| | Yes | 280 | 57.9 |
| Types of contraceptive use (n = 280) | Injection | 142 | 50.7 |
| | Implants | 105 | 37.6 |
| | Pill | 24 | 8.6 |
| | Others[a] | 9 | 3.1 |
| Place of contraceptive use(n = 280) | Health post | 22 | 7.9 |
| | Health center | 167 | 59.6 |
| | Hospital | 56 | 20 |
| | Private clinic/Pharmacy shop | 35 | 12.5 |
| Reason for not use contraception (n = 204) | Side effect | 51 | 25 |
| | No availability | 15 | 7.4 |
| | Lack of information | 23 | 11.8 |
| | Opposition from, husband & religious leader | 16 | 7.8 |
| | Have desire for pregnancy | 98 | 48 |
| Intake of Iron-folic acid during the current pregnancy | No | 309 | 40.4 |
| | Yes | 455 | 59.6 |
| Currently, intake of coffee (n = 584) | No | 92 | 15.8 |
| | Yes | 492 | 84.2 |
| Fasting (abstaining from animal-source foods) | No | 112 | 14.7 |
| | Yes | 652 | 85.3 |
| Vaccination status | Td1 | 555 | 72.6 |
| | Td2 | 470 | 61.5 |
| | Td3 | 328 | 42.9 |
| | Td4 and above | 266 | 34.8 |
| | Covid-19 | 665 | 87 |
| | Hepatitis-B | 48 | 6.3 |
| | HPV | 36 | 4.7 |
| Test HIV for the current pregnancy | No | 247 | 32.3 |
| | Yes | 517 | 67.7 |
| Test for hepatitis B for the current pregnancy | No | 550 | 72 |
| | Yes | 214 | 28 |
| Screen for cervical cancer at least once within the past 5 years | No | 742 | 97.1 |
| | Yes | 22 | 2.9 |
| Did you screen for anemia during your current pregnancy | No | 448 | 58.6 |
| | Yes | 316 | 41.4 |

NB: [a]Others (Emergency contraceptive, Breastfeeding & Male condom)

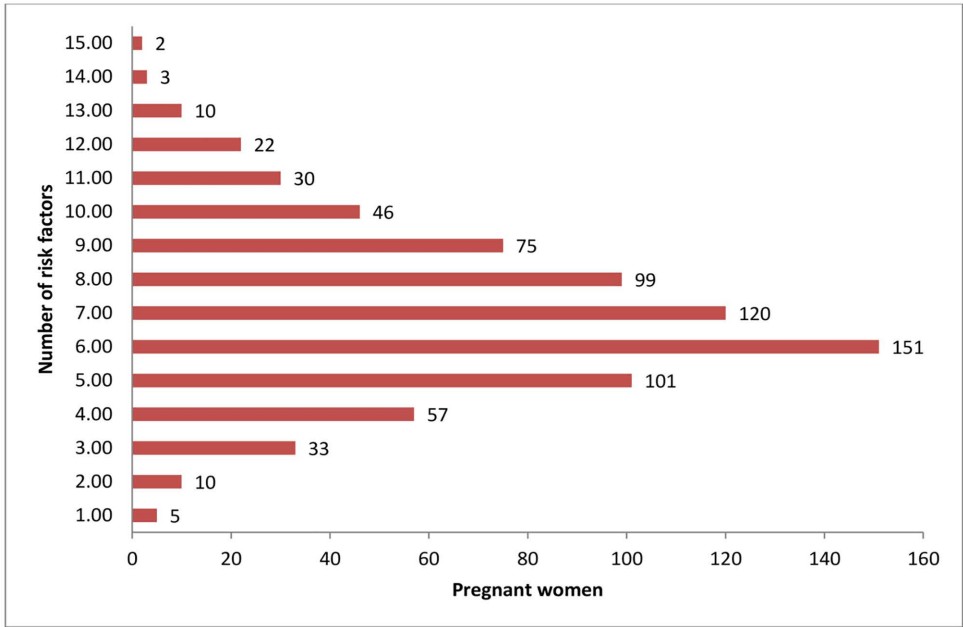

**Fig 4. Distribution of overall risk factors for poor pregnancy outcomes among participants in Tigray, Ethiopia, 2024 (n = 764).**

education, and promote better family planning and pregnancy preparation by encouraging timely medical intervention. This calls for the improvement of the quality of obstetrics service as one effort to improve the maternal and fetal outcomes.

Evidence suggests that most individuals at risk before conception continue to experience these risks during pregnancy, with risk factors being more prevalent in the preconception stage. This highlights the need for innovative strategies to enhance prevention efforts [79]. In this study, all pregnant women were exposed to at least one risk factor for poor pregnancy outcomes, with 71.5% facing 5–9 types.

This is notably higher than a U.S. study, where only 32% of pregnant women faced one or more of three risk factors [79]. The disparity may be due to differences in healthcare access, as U.S. women average 6.4 physician visits annually, allowing more opportunities to address risks [79]. Other factors include variations in sample size, risk measurement, socioeconomic status, healthcare-seeking behavior, health systems, and education levels. On the other hand, studies in India found that all participants had at least one risk factor, with over 70% reporting five or more [3], aligning with our findings. Emphasizing the identification of risk factors for APOs from the preconception period is essential for integrating services into the maternal continuum of care.

The health belief model, rooted in Rosenstock's psychological and behavioral theory, suggests that individuals are more likely to adopt health-related behaviors when they perceive a significant risk of negative consequences for not taking action. The model highlights that personal motivation to act arises from a sense of threat, with key predictors such as perceived susceptibility to risks, which is a subjective evaluation of the severity of the disease, playing a vital role in shaping behaviors like screenings and counseling [80]. Consequently, recognizing one's susceptibility to preconception risks is essential for increasing the uptake of PCC services. The study found that participants with high and moderate perceived susceptibility to preconception risks were more likely to utilize PCC services than those with low susceptibility. Therefore, implementing a strategy that provides routine services to all women of reproductive age, using a reproductive life plan tool, can help healthcare providers identify eligible women and assess preconception risk burdens, ultimately enhancing service-seeking behavior. HIV screening serves as a crucial gateway to PCC by increasing awareness, fostering trust, and promoting proactive health behaviors, ultimately leading to healthier pregnancy outcomes.

**Table 5. Risk factors for adverse pregnancy outcomes among participants in Tigray, Ethiopia, 2024(n = 764).**

| Variables | | Frequency | Percentage |
|---|---|---|---|
| Intake alcohol during current pregnancy | No | 326 | 42.7 |
| | Yes | 438 | 57.3 |
| Amount of intake of coffee per day | 1-2 cups per day | 402 | 52.6 |
| | ≥3 cups per day | 362 | 47.4 |
| History of smoking[a] | No | 750 | 98.2 |
| | Yes | 14 | 1.8 |
| Current medication use | No | 572 | 74.9 |
| | Yes | 192 | 25.1 |
| Dietary diversity level | Non adequate | 544 | 71.2 |
| | Adequate | 220 | 28.8 |
| Physical activities | Active | 267 | 34.9 |
| | Inactive | 497 | 65.1 |
| MUAC[b] | Well nourished | 386 | 50.5 |
| | Undernourished | 378 | 49.5 |
| Infertility/sub-infertility | No | 729 | 95.4 |
| | Yes | 35 | 4.6 |
| Pregnancy interval | Normal PI[c] | 458 | 59.9 |
| | Abnormal PI | 306 | 40.1 |
| Plan pregnancy | Planned | 316 | 41.4 |
| | Unplanned | 44 8 | 58.6 |
| History of APOs | No | 383 | 71.2 |
| | Yes | 220 | 28.8 |
| Grand multipara | No | 693 | 90.7 |
| | Yes | 71 | 9.3 |
| Maternal age risks | No | 574 | 75.1 |
| | Yes | 190 | 24.9 |
| History of Medical conditions | No | 625 | 81.8 |
| | Yes | 139 | 18.2 |
| Predental health problem | No | 595 | 77.9 |
| | Yes | 169 | 22.1 |
| Psychological distress | No | 672 | 88 |
| | Yes | 92 | 12 |
| Environmental & household exposure | No | 6 | 0.8 |
| | Yes | 758 | 99.2 |
| Partner violence | No | 747 | 97.8 |
| | Yes | 17 | 2.2 |
| Have genetic risks | No | 687 | 89.9 |
| | Yes | 77 | 10.1 |
| Have anorexia | No | 495 | 64.8 |
| | Yes | 269 | 35.2 |
| Insufficient sleep (average less than 6 hours per night) | No | 653 | 85.5 |
| | Yes | 111 | 14.5 |

**NB:** [d]smoking history, including both active smoking and exposure for smoking; [b]MUAC, Middle upper circumference; [c]PI,pregnancy interval; [d]APOs, Adverse pregnancy outcomes.

**Table 6. Bivariate and multivariable logistic regression analysis of factors associated with the uptake of PCC services among participants in Tigray, Ethiopia, 2024(n = 764).**

| Variables | Uptake of PCC | | Crude Odds Ratio | | Adjusted Odds Ratio | |
|---|---|---|---|---|---|---|
| | Yes,n(%) | No,n(%) | Odds Ratio | 95% CI | Odds Ratio | 95% CI |
| Women's decision-making power | | | | | | |
| No | 19 (3.6%) | 521 (96.4%) | 1.00 | | 1.00 | |
| Yes | 36 (16.1%) | 188 (83.9%) | 1.77 | 1.39–2.27 | 5.27 | 2.85–9.72* |
| Residence | | | | | | |
| Rural | 28 (5.4%) | 487 (94.6%) | 1.00 | | 1.00 | |
| Urban | 27 (10.8%) | 222 (89.2%) | 2.1 | 1.22–3.67 | 1.56 | 0.78–3.11 |
| Husband occupation | | | | | | |
| Farmer | 10 (4.2%) | 228 (95.8%) | 1.00 | | 1.00 | |
| Self employed | 19 (11.4%) | 147 (88.6%) | 2.9 | 1.3–6.5 | 1.929 | 0.75–4.98 |
| Government employed | 11 (12.4%) | 78 (87.6%) | 3.2 | 1.3–7.8 | 2.628 | 0.9–7.67 |
| Daily laborer | 11 (6%) | 172 (94%) | 1.46 | 0.6–3.5 | 1.412 | 0.52–3.81 |
| No work | 4 (4.5%) | 84 (95.5%) | 1.06 | 0.33–3.55 | 0.935 | 0.26–3.34 |
| History of medical illness | | | | | | |
| No | 50 (8%) | 575 (92%) | 1.00 | | 1.00 | |
| Yes | 5 (3.6%)) | 134 (96.4%) | 0.43 | 0.17–1.10 | 0.43 | 0.16–1.17 |
| Planned pregnancy | | | | | | |
| Unplanned | 28 (6.3%) | 420 (93.8%) | 1.00 | | 1.00 | |
| Planned | 27 (8.5%) | 289 (91.%) | 1.4 | 0.81–2.43 | 1.44 | 0.78–2.64 |
| Information about PCC | | | | | | |
| No | 40 (6%) | 631 (94%) | 1.00 | | 1.00 | |
| Yes | 15 (16.1%) | 78 (83.9%) | 3.03 | 1.6–5.74 | 2.9 | 1.42–5.91* |
| Member of community health insurance | | | | | | |
| No | 47 (6.8%) | 644 (93.2%) | 1.00 | | 1.00 | |
| Yes | 8 (11%) | 65 (89%) | 1.98 | 0.76–3.72 | 2.10 | 0.85–5.16 |
| Tested HIV for the current pregnancy | | | | | | |
| No | 8 (3.2%) | 239 (96.8%) | 1.00 | | 1.00 | |
| Yes | 47 (9.1%) | 470 (90.9%) | 2.98 | 1.39–6.42 | 2.27 | 1.01–5.11* |
| Perceived susceptibility to preconception health risks | | | | | | |
| Low | 5 (2.6%) | 191 (97.4%) | 1.00 | | 1.00 | |
| Moderate | 19 (9.5%) | 180 (90.5%) | 4.03 | 1.47–11.03 | 4.38 | 1.53–12.52* |
| High | 31 (8.4%) | 338 (91.6%) | 3.5 | 1.34–9.16 | 3.25 | 1.19–8.85* |

NB: *P=value less than 0.05.

Additionally, frequent interactions with healthcare professionals enhance access to PCC information, likely due to improved health-seeking behaviors [81]. Our study found that women who underwent HIV screening before pregnancy were more likely to utilize PCC services than those who did not. By integrating PCC into HIV screening settings, healthcare providers can leverage this opportunity to educate and support women in optimizing their reproductive health before conception.

Evidence suggests that women's decision-making power enhances maternal health service utilization [82] and improves access to PCC services [83]. In this study, women with greater decision-making power were 5.27 times more likely to utilize PCC compared to their counterparts. This may be attributed to their increased ability to seek information, make informed

decisions about their reproductive health, including family planning and optimal pregnancy timing, and actively participate in overcoming social, cultural, and economic barriers that hinder access to essential healthcare services.

## Limitations and strengths of the study

The strength of this study lies in collecting information from pregnant women in their first and second trimesters, which helps minimize recall bias. Moreover, the inclusion of participants from both urban and rural settings enhances the study's inclusivity. However, the reliance on self-reported data still introduces potential limitations, such as social desirability bias, which may impact the validity and reliability of the results. Another limitation is the use of a 24-hour dietary recall to assess dietary diversity in pregnant women. Although it is an internationally recommended and feasible tool for population-level analysis, it relies on memory and may not capture habitual individual intake due to daily variations. Additionally, we did not perform validity testing for the attitude measurement.

Additionally, as the data was collected solely from women, a couple-based approach would have been more helpful to better understand the factors influencing PCC uptake among pregnant women from the study communities.

## Conculsion

The uptake of PCC services among pregnant women in this study was notably low, despite a high prevalence of risk factors for APOs. Women's decision-making power, HIV testing during the current pregnancy, knowledge about PCC, and perceived susceptibility to preconception risks were statistically significant factors associated with PCC service uptake. To improve PCC uptake, integrated interventions should: (1) implement targeted education to enhance PCC awareness and risk perception; (2) incorporate PCC counseling into existing services like HIV testing and antenatal care; (3) leverage community health platforms (e.g., Health Extension Program) to identify eligible women; and (4) train providers to use tools like the Reproductive Life Plan for routine preconception risk screening. Future studies should assess PCC uptake among high-risk women, and couples alongside targeted interventions to enhance the uptake of PCC services. Additionally, comprehensive preconception risk assessments are essential for accurately measuring the true burden of risks, enhancing participants perceived susceptibility, which in turn helps with seeking behaviors, guides healthcare priorities, and informs the planning of effective interventions.

## Supporting information

**S1 Dataset. Supporting information, dataset in SPSS on the uptake of preconception care.**
(SAV)

**S2 Dataset. Supporting information, dataset in Excel on the uptake of preconception care.**
(CSV)

**S1 Appendix. Questionnaire about uptake of preconception care.**
(DOCX)

## Acknowledgments

We sincerely acknowledge Mekelle University, the Tigray Regional Health Bureau, and JSI Ethiopia for their financial support. We also extend our gratitude to the supervisors and data collectors for their dedication and to the study participants for their voluntary participation.

## Author contributions

**Conceptualization:** Gebremedhin Gebreegziabher Gebretsadik, Andargachew kassa Biratu, Alemayehu Bayray Kahsay, Amanuel Gessessew, Hailemariam Segni, Zohra S. Lassi, Afework Mulugeta.

**Data curation:** Gebremedhin Gebreegziabher Gebretsadik.

**Formal analysis:** Gebremedhin Gebreegziabher Gebretsadik.

**Investigation:** Gebremedhin Gebreegziabher Gebretsadik.

**Methodology:** Gebremedhin Gebreegziabher Gebretsadik, Andargachew kassa Biratu, Alemayehu Bayray Kahsay, Amanuel Gessessew, Hailemariam Segni, Zohra S. Lassi, Afework Mulugeta.

**Resources:** Gebremedhin Gebreegziabher Gebretsadik.

**Software:** Gebremedhin Gebreegziabher Gebretsadik.

**Supervision:** Gebremedhin Gebreegziabher Gebretsadik, Andargachew kassa Biratu, Alemayehu Bayray Kahsay, Amanuel Gessessew, Hailemariam Segni, Afework Mulugeta.

**Validation:** Gebremedhin Gebreegziabher Gebretsadik, Zohra S. Lassi, Afework Mulugeta.

**Visualization:** Gebremedhin Gebreegziabher Gebretsadik.

**Writing – original draft:** Gebremedhin Gebreegziabher Gebretsadik.

**Writing – review & editing:** Gebremedhin Gebreegziabher Gebretsadik, Andargachew kassa Biratu, Alemayehu Bayray Kahsay, Amanuel Gessessew, Hailemariam Segni, Zohra S. Lassi, Afework Mulugeta.

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
