## [Decision Letter · Decision Letter 0]

28 Aug 2025

Dear Dr. Gebretsadik,

Thank you for submitting your manuscript to PLOS ONE. After careful consideration, we feel that it has merit but does not fully meet PLOS ONE’s publication criteria as it currently stands. Therefore, we invite you to submit a revised version of the manuscript that addresses the points raised during the review process.

We look forward to receiving your revised manuscript.

Kind regards,

Kahsu Gebrekidan, Ph.D.

Academic Editor

PLOS ONE

Journal Requirements: 

2. Please amend either the abstract on the online submission form (via Edit Submission) or the abstract in the manuscript so that they are identical.

5.  We are unable to open your Supporting Information file [PCC data_3.sav]. Please kindly revise as necessary and re-upload.

6. In the ethics statement in the Methods, you have specified that verbal consent was obtained. Please provide additional details regarding how this consent was documented and witnessed, and state whether this was approved by the IRB.

Reviewers' comments:

Reviewer's Responses to Questions

**Comments to the Author**

1. Is the manuscript technically sound, and do the data support the conclusions?

Reviewer #1: Yes

Reviewer #2: Yes

2. Has the statistical analysis been performed appropriately and rigorously?

Reviewer #1: Yes

Reviewer #2: Yes

3. Have the authors made all data underlying the findings in their manuscript fully available?

Reviewer #1: Yes

Reviewer #2: Yes

4. Is the manuscript presented in an intelligible fashion and written in standard English?

Reviewer #1: Yes

Reviewer #2: Yes

Reviewer #1: Thank you for this invitation. Please see my comments as follows:

Abstract:

1. The method section needs more details such as how participants selected using interviews? And authors did not write that they conducted the study among pregnant or non-pregnant women. The instruments and outcomes are so important, and authors should mention these two factors.

2. The participants were in which gestational age, and were they primipara or multipara?

3. What authors mean about PCC service uptake was 7.2%? Is that mean that only 7.2% of women used PCC?

4. Please write the conclusion according to your results. In this study you did not evaluate the knowledge. Only focus on your objectives.

Introduction

1. What is HSTP II? Please write each word in full before using its abbreviation.

2. Please write about the current situation of PCC in Ethiopia? Is PCC mandatory for women in reproductive age? and if so, who is responsible to do this care in public health centers?

3. How many visits are considered as optimal for PCC?

4. Is PCC active in your region? For instance, if the woman dose not come to the health center, a health provider, will call them?

Methods

1. What is MUAC ?

2. What is the justification for 778 of sample size?

3. Measurements of PCC service uptake: please clarify for readers, if women received PCC in few fragments, for instance, in one session they received only screening or counselling, or if they receive PCC in a single package.

4. How authors evaluated the validity and reliability of instrument that they used for assessing attitude?

5. Line 190: While authors intended to assess the PCC, why they asked about ultrasound service within the first 24 weeks of gestation?

6. Line 208: What do you mean about street medications? Do you mean home-remedies?

7. Line 214: Did you evaluate alcohol drinking only during pregnancy or ever consumption?

8. Line 230: Please provide reference for the mid-upper arm circumference of > 23 indicated well nutritional status.

9. In my opinion, for assessing dietary diversity, authors should have used a better questionnaire, not just asking participants about what they consumed during the past 24 hours.

10. What were the inclusion/exclusion criteria?

11. How did you classify the attitude to poor and good?

Results

1. Table 1: In my opinion, because many people nowadays follow health instruction via their smart phone, they do not need to have TV or radio. So, it was better that authors asked about smart phone.

2. Line 334: What do you mean about “women reported fasting during their pregnancy”? Is it due to the religious rules or just for diet?

3. Table 5; Please explain about physical activity and how did you divide participants into active or inactive?

Reviewer #2: Comments for Authors

Abstract

Comments on the Title:

1. The title should clearly reflect the study’s focus on preconception care (PCC) uptake and pregnant women's experiences in Tigray, Ethiopia.

2. Consider including key elements such as:

* The study design(e.g.community-based cross-sectional study”)

* Study population (pregnant women)

* Location and time frame (Tigray, 2024)

3.A more informative title could be:

Preconception Care Uptake and Associated Factors Among Pregnant Women in Tigray, Ethiopia: A Community-Based Cross-Sectional Study

Methodology

1. Consider updating the antenatal care statistics with more recent data if available, as the current reference is from the 2007 census

2.Good justification is given for trimester selection, but clarify whether third-trimester women were excluded and why?

3.The reliance on HEWs and WDGs may introduce selection bias; briefly acknowledge or address this potential limitation.

4. Report how the final sample size (764) was calculated or adjusted from the initial 778, and whether any exclusions occurred during data collection.

5. Mention any pilot testing or pretesting of the questionnaire (even if briefly) to validate tools for local context.

6.In Access to Health Services: Clarify if any adjustments were made for terrain, transportation availability, or security issues, especially in post-conflict settings.

7.The PCC components list is comprehensive, but you might consider summarizing or using a table for readability.

**Do you want your identity to be public for this peer review?** For information about this choice, including consent withdrawal, please see our Privacy Policy

Reviewer #1: **Yes: ** Prof Parvin Abedi

Reviewer #2: No

---

## [Author Response · Author response to Decision Letter 1]

30 Sep 2025

Thank you, Editor and Reviewers, for your valuable comments and suggestions. We have addressed each comment individually in the response to reviewers document. We hope the revised manuscript will be considered for publication.

---

## [Decision Letter · Decision Letter 1]

11 Oct 2025

Dear Dr.  Gebretsadik,

Thank you for submitting your manuscript to PLOS ONE. After careful consideration, we feel that it has merit but does not fully meet PLOS ONE’s publication criteria as it currently stands. Therefore, we invite you to submit a revised version of the manuscript that addresses the points raised during the review process.

We look forward to receiving your revised manuscript.

Kind regards,

Kahsu Gebrekidan, Ph.D.

Academic Editor

PLOS ONE

Journal Requirements:

Reviewer's Responses to Questions

**Comments to the Author**

Reviewer #1: All comments have been addressed

Reviewer #2: All comments have been addressed

2. Is the manuscript technically sound, and do the data support the conclusions?

Reviewer #1: Yes

Reviewer #2: Yes

3. Has the statistical analysis been performed appropriately and rigorously?

Reviewer #1: Yes

Reviewer #2: Yes

4. Have the authors made all data underlying the findings in their manuscript fully available?

Reviewer #1: Yes

Reviewer #2: Yes

5. Is the manuscript presented in an intelligible fashion and written in standard English?

Reviewer #1: Yes

Reviewer #2: Yes

Reviewer #1: Although authors added some information about how they assessed the validity and reliability of instrument for measuring attitude, their answer was not convincing. For instance, they should have report at least CVI, and CVR and Cronbach's alpha.

Reviewer #2: All my comments has been addressed accordingly. But i wanna add some comments as follow:

1.This abstract introduction effectively highlights a critical global health issue—poor pregnancy outcomes—with a specific focus on low- and middle-income countries. It correctly emphasizes the importance of preconception care (PCC) as a preventive strategy, but also points out a key gap in both implementation and research, especially in conflict-affected regions. The choice to focus on Tigray, Ethiopia, adds relevance and urgency given the area's recent instability. To strengthen this paragraph, the authors might consider briefly defining what PCC includes, or providing a statistic or reference to underscore the magnitude of poor pregnancy outcomes globally. Overall, the paragraph sets a clear and important context for the study.

**Do you want your identity to be public for this peer review?** For information about this choice, including consent withdrawal, please see our Privacy Policy

Reviewer #1: **Yes: ** Prof Parvin Abedi

Reviewer #2: No

---

## [Author Response · Author response to Decision Letter 2]

17 Oct 2025

We have submitted a detailed, line-by-line response to the editors' and reviewers' comments in the document "Response to Reviewers Two." All raised issues have been thoroughly addressed, and we hope this leads to acceptance of our manuscript for publication.

---

## [Editor Report · Decision Letter 2]

23 Oct 2025

Preconception Care Uptake and Risk Factors for Adverse Pregnancy Outcomes among pregnant women in Tigray, northern Ethiopia: A Community-Based Cross-Sectional Study

PONE-D-25-35031R2

Dear Dr. Gebremedhin,

We’re pleased to inform you that your manuscript has been judged scientifically suitable for publication and will be formally accepted for publication once it meets all outstanding technical requirements.

Kind regards,

Kahsu Gebrekidan, Ph.D.

Academic Editor

PLOS ONE
---

## [Editor Report · Acceptance letter]

PONE-D-25-35031R2

PLOS ONE

Dear Dr. Gebretsadik,

I'm pleased to inform you that your manuscript has been deemed suitable for publication in PLOS ONE. Congratulations! Your manuscript is now being handed over to our production team.

Kind regards,

on behalf of

Dr. Kahsu Gebrekidan

Academic Editor

PLOS O